# Validation of an Intra-Oral Scan Method Versus Cone Beam Computed Tomography Superimposition to Assess the Accuracy between Planned and Achieved Dental Implants: A Randomized In Vitro Study

**DOI:** 10.3390/ijerph17249358

**Published:** 2020-12-14

**Authors:** Alessio Franchina, Luigi V. Stefanelli, Fabio Maltese, George A. Mandelaris, Alessandro Vantaggiato, Michele Pagliarulo, Nicola Pranno, Edoardo Brauner, Francesca De Angelis, Stefano Di Carlo

**Affiliations:** 1Private Practice, Periodontics and Dental Implant Surgery, 36100 Vicenza, Italy; alessiofranchina@icloud.com; 2Department of Oral and Maxillo-Facial Sciences, Sapienza University of Rome, 00185 Rome, Italy; gigstef@libero.it (L.V.S.); nicola.pranno@uniroma1.it (N.P.); edoardo.brauner@uniroma1.it (E.B.); stefano.dicarlo@uniroma1.it (S.D.C.); 3Private Practice, 00192 Rome, Italy; drfabiomaltese@gmail.com; 4Private Practice, Periodontics and Dental Implant Surgery, Periodontal Medicine & Surgical Specialists, LTD, Park Ridge, Oakbrook Terrace, Chicago, IL 60601, USA; gmandelaris@periodontalmedicine.org; 5Private Practice, 73100 Lecce, Italy; alessandrovantaggiato@gmail.com; 6Faculty of Dental Medicine, University of Plovdiv, 4002 Plovdiv, Bulgaria; michelepagliarulo2000@gmail.com

**Keywords:** accuracy, superimposition, intra-oral scanning, surgical guides, static computer aided implantology, dynamic computer aided implantology, dynamic navigation implantology

## Abstract

Computer aided implantology is the safest way to perform dental implants. The research of high accuracy represents a daily effort. The validated method to assess the accuracy of placed dental implants is the superimposition of a pre-operative and a post-operative cone beam computed tomography (CBCT) with planned and placed implants. This procedure is accountable for a biologic cost for the patient. To investigate alternative procedure for accuracy assessment, fifteen resin casts were printed. For each model, six implants were digitally planned and then placed following three different approaches: (a) template guided free hand, (b) static computer aided implantology (SCAI), and (c) dynamic computer aided implantology (DCAI). The placement accuracy of each implant was performed via two methods: the CBCT comparison described above and a matching between implant positions recovered from the original surgical plan with those obtained with a post-operative intraoral scan (IOS). Statistically significant mean differences between guided groups (SCAI and DCAI) and the free hand group were found at all considered deviations, while no differences resulted between the SCAI and DCAI approaches. Moreover, no mean statistically significant differences were found between CBCT and IOS assessment, confirming the validity of this new method.

## 1. Introduction

Dental implants have dramatically changed the outcome of oral health conditions of patients with missing teeth [1,2,3]. To make a complete study of the patient’s jaws, radiographic examinations are mandatory. The 2D digital X-rays give limited information, especially when the operator is investigating the jaws’ width or the proximity to neighboring relevant structure as nerves or roots [4].

A great advancement in this diagnostic field has been reached with the introduction of the 3D volumetric X-ray. With these devices, the clinician is able to carry out a deep examination of the bone area where the implant treatment has to be performed with a reasonable low exposure to radiation [5,6,7].

The insertion of the implants in an optimal position is an important key to reach a long-term success rate of implant and prosthetic rehabilitation’s stability. Moreover, the lack of compliance to the proper implant plan in addition to midfacial thin soft tissues has been identified as a medium–long term cause of biologic complications such as mucositis and perimplantitis [8,9,10,11].

To avoid these issues, technologies have been developed to offer the clinician several options for more predictable results. The latest digital solutions allow guiding the implant site drilling as well as the implant seating, resulting in less prosthetic compromises and greater implant placement accuracy [12].

The literature reports two different ways of carrying out this procedure safely. Using static guided surgery (SCAI, static computer aided implantology), a surgical stent is digitally designed using Digital Imaging and Communications in Medicine (DICOM) data obtained with 3D X-ray and then prototyped by using a 3D printer. The implant site is prepared to seat the implant in the planned position thanks to a dedicated surgical kit. The second approach is dynamic guided surgery (DCAI, dynamic computer aided implantology), a method allowing a real-time check of the surgical burs 3D position during the osteotomy related to the planned implant [13].

The existing discrepancies between a planned implant and an inserted implant are one of the most relevant challenges for the operators to understand if the method used for that surgery is accurate enough to guarantee the requested results [14,15,16].

Up to now, the method validated by the literature to calculate this accuracy is the superimposition of the post-operative CBCT of the treated jaw (with the inserted implants) with the pre-operative jaw (with the planned implants) [17,18].

However, the CBCT scan method involves a biological cost for the patients.

All these comparison methods use an iterative software that runs until the best fitting between the jaw in the pre-operative images and that one of the post-operative images has been found.

After that, it is easy to automatically calculate the accuracy by using meshes’ software, which allows for the identification of the 3D coordinates at the entry and apex center of the planned and inserted implants [14].

There are few in vitro studies reported by the literature, validating the use of the impression taken with an Intra Oral Scanner (IOS), as previously described for the use of pre-op and post-op CBCT’s superimposition [19,20].

This last procedure, which has the benefit of avoiding a second CBCT scan for the patient, could become routine, if a study validating its accuracy is carried out.

The aim of this study was to find out these results, making a comparison between the two above-mentioned methods (IOS vs. CBCT).

The accuracy of implant insertion by using free-hand, static guide, and a dynamic navigation system was also evaluated in terms of intra- and inter-operator deviations.

The null hypothesis is the lack of a statistically significant difference by using the two CAI methods and that a CAI (i.e., static or dynamic) approach is more accurate than a free hand way.

## 2. Materials and Methods

The study was designed to evaluate the in vitro accuracy of implant insertion by using three different approaches: free-hand (template guided), static guided (Naviguide^®^, Biomax Spa, Vicenza, VI, Italy), and dynamic navigation system (Navident^®^, ClaroNav, Toronto, ON, Canada).

Starting from the DICOM data of a real clinical case, a sample of fifteen resin models was printed (Form3, Formlabs) from a dental laboratory and six implants per cast were planned and placed by using the three above-mentioned modalities.

The cast reproduced a lower jaw with some residual teeth and two partially edentulous spans, where implants were planned and placed (Figure 1).

Free hand and DCAI surgeries were carried out by a tapered surgical kit (ZimmerBiomet^®^, Warsaw, IN, USA), following the manufacturer’s recommendations. The SCAI surgeries were carried out using a dedicated trousse (ZimmerBiomet Navigator^®^) following the above-mentioned surgical plan. A number of six implants per model (ZimmerBiomet Certain^®^ 4.1 mm × 13 mm) were planned, three implants per side.

Two implants per cast were placed free hand (4.6 and 3.5), two implants with a static computer-assisted implant guide (4.7 and 3.4), and the last two implants (4.5 and 3.6) with a dynamic computer-assisted system.

Both of the implants’ plan and surgeries were performed by A.F., L.V.S., and F.M. in the same office, on the same day and with the same devices. The operators were general dentists, perfectioned in the field of implantology, with an average number of placed implants per year higher than 250 implants.

The allocation of the casts as well as the treatment modality (FH, SG, ND) were randomized for each implant site. In addition, the timing of treatment modality was also randomized.

To perform this analysis two methods were used:A comparison between pre-op and post-op CBCT, based on radiographic volume superimposition;A comparison between post-op Standard Tessellation Language (STL) files obtained from an intra-oral scan (IOS) of the resin casts and the STL file obtained from the pre-op CBCT implant planning. In this case, the remaining teeth were used to align pre-op and post-op STL files to assess implants deviations.

### 2.1. Static Computer Aided Implantology Workflow (SG)

The Navibox^®^ SCAI workflow is based on the merge of the prosthetic project (represented by an STL file of the wax-up, that can be performed digitally or analogically), the CBCT DICOM files of the jaw where the rehabilitation is needed, and the actual condition (soft tissue and teeth). In this system, the CBCT has to be taken using a plastic device (Navibite^®^) containing a radiopaque fiducial marker useful for the above-mentioned overlapping step (Figure 2). This basic and crucial operation was carried out by the manufacturer only.

Once the merging step had been carried out by the company, the 3D models of the actual oral status and wax-up were visible (Figure 3) in the 3D window of the surgical software (Navimax^®^, Biomax Spa, Vicenza, VI, Italy).

The implant planning was prosthetically driven (Figure 4) by the wax-up image in the CBCT cross sections, following the guidelines previously described [21]. Once the implant planning was performed, the guide order was forwarded.

To perform this study, three identical surgical guides (Figure 5) were printed (Form3, Formlabs) using specific PEEK (Polieter-eter-keton) sleeves embedded. Each sleeve (Figure 5) can be planned at 5 mm or 8 mm distance from the implant shoulder (i.e., implant connection), depending on the case at hand. This means that the specific surgical kit includes two bur lengths, short or long.

The surgical guides were provided with a surgical plan including the drill to be used to comply with the planning.

### 2.2. Dynamic Computer Aided Implantology Workflow (ND)

Dynamic Navigation workflow unfolds in three steps: planning, tracing, and placing.

#### 2.2.1. Plan

In this case, the DICOM (Digital Imaging Communication in Medicine) of a CBCT (Scanora 3Dx) was uploaded into Navident software that rendered them in five quadrants: a 3D, a panorex, an axial, a sagittal, and a coronal view (Figure 6).

After importing the DICOM files, a second import of intraoral surface scan IOS (CS3600, Carestream Dental LLC) took place. The two imported files were then matched using a minimum of three markers on both datasets.

Implant placement was then prosthetically planned (Figure 6), optimizing both ideal positions according to diagnostic digital wax up and bone availability.

#### 2.2.2. Trace

As said, using a contact scanner (Tracer Tool by Navident) to trace between 3 and 6 radiopaque crowns allows software to match crown 3D mesh with crown 3D DICOM rendering, achieving CBCT registration to the patient’s anatomy.

In order to perform “tracing”, it is necessary to fix on a patient a black and white checkerboard HeadTracker™ if maxilla, or JawTracker™ if mandible (ClaroNav™, Toronto, ON, Canada) (Figure 7). Equally, the tracer tool (Tracer™, ClaroNav™, Toronto, ON, Canada) will mount a checkerboard. Both were detected by MicronTracker (ClaroNav™, Toronto, ON, Canada) motion tracking cameras and software will superimpose these pieces of information onto the CBCT, allowing for the accomplishment of the “tracing” stage.

In order to prove the accuracy of the procedure, an “Accuracy Test” needs to be carried out. This consists in touching any crown cusps with the tracer and checking on the laptop screen if the same cusp has been touched. If this is the case, then the test is positive and the surgeon can move on with the protocol. In case of incongruency, the whole process should be repeated until a positive test is achieved.

#### 2.2.3. Place

A metallic caliber was used to communicate with the software. The caliber, depending on the fixed position, allows to calibrate the tracers, contrangle axis, drill length, implant length, piezo tips, and saws. Soon before surgery, a contrangle chuck gets inserted on a caliber peg and twisted for a quarter of a circle. By doing this, the specific contrangle angle is calibrated. Soon after, the pilot drill gets inserted on a dimple numbered 2 to be held in position for a few seconds and therefore gets calibrated. Once finished, a new “accuracy test” needs to be carried out and a positive result allows the surgery to start.

Getting close to the planned implant, a crosshair shows up and the surgeon should maintain drilling at the center of this target until the planned implant gets to the apex (Figure 8).

### 2.3. Free Hand Workflow (FH)

Two implants per model were planned, prepared, and seated free hand.

In order to obtain an ideal implant placement and to comply with the prosthetic requests, a surgical template was considered useful. The diagnostic phase of the sample of implant sites was performed in the same manner described for SCAI sites. For this reason, the surgical guides were designed leaving the shape of the wax to give to the guide that final outer contour (Figure 9). After the guides were printed, only the occlusal surface of each free hand site was trimmed with a bur, leaving intact all the surrounding walls.

This procedure allowed the operators to control the 3D direction of the burs during the drilling step, making the site preparation suitable.

### 2.4. Digital Data Collection

After the surgeries, the fifteen models with implants placed were scanned with CBCT (Soredex Scanora 3D CBCT, Kavo Kerr, Brea, CA, USA) using the same scan settings and parameters (14-bit gray density, 0.25 mm voxel size, 90 KV) as well as with an IOS device (CS3600, Carestream Dental LLC, Atlanta, GA, USA).

A dedicated scan body (Biomax spa, Italy) was screwed on each implant before taking the digital impression in order to create, in a second stage and with a dedicated CAD software (Exocad^®^GmbH, Darmstadt, Germany), a mesh (an STL file) containing the placed implants.

After that, all measurements were performed using industrial software (Mimics, Materialise, Leuven, Belgium) by a superimposition of the STL files containing the planned implants, first with the STL files of the achieved implants (both derived from the CBCT DICOM data) as the control group, and second, with the STL files derived from the IOS of the placed implants as a test group (Figure 10 and Figure 11).

Finally, the spatial coordinates of the planned and inserted implants at the center of the entry point and apex point were reported in an Excel spreadsheet and the deviations were automatically calculated in terms of coronal deviations (mm), apical deviations (mm), angular deviations (degree), and depth (apical) deviations (mm).

## 3. Statistical Analysis

An Excel spreadsheet was used to create a dataset. Mean values were reported for each variable and data outlier were described by box plots. To assess if the data complied with a normal distribution, a Shapiro–Wilk test was used.

To identify whether a statistically significant mean difference existed in the three different techniques used to insert the implants, a one-way Analysis of Variance (ANOVA) was used. Games–Howell post-hoc analysis was performed for multiple comparisons.

The independent-samples t-test was used to identify the statistically significant mean difference using a post-operative CBCT or a post-operative IOS impression for the implant accuracy calculation. Data were evaluated using standard statistical analysis software (SPSS version 20.0, Statistical Package for the Social Sciences, IBM Corporation, Armonk, NY, USA). A *p* ≤ 0.05 cut off was used for statistical significance.

## 4. Results

Ninety implants were inserted, 30 implants free-hand, 30 implants with a static guide, and 30 implants with the dynamic navigation system.

The data, in terms of accuracy by using the three different techniques, are reported in Table 1 and Figure 12.

At coronal, apical, and depth level, the best result was exhibited by the **SG** approach (mean coronal deviation 0.79 mm ± 0.35 mm, mean apical deviation 1.17 mm ± 0.48 mm, mean depth deviation 0.36 mm ± 0.29 mm), followed by the **ND** (mean coronal deviation 0.89 mm ± 0.37 mm, mean apical deviation 1.31 mm ± 0.68 mm, mean depth deviation 0.51 mm ± 0.46 mm), and the **FH** approach (mean coronal deviation 1.65 mm ± 0.61 mm, mean apical deviation 2.33 mm ± 1.01 mm, mean depth deviation 0.83 mm ± 0.49 mm).

The angular results showed the best performance with the **ND** approach (2.76° ± 0.61°), followed by the **SG** method (3.23° ± 1.00°) and the **FH** method (7.41° ± 3.87°).

No statistically significant differences between the means of deviations were found by using the two CAI methods. Furthermore, there was a statistically significant difference between the means of a CAI method (static or dynamic) vs. free hand in all the considered variables (Figure 13), as illustrated by the data in Table 2.

Two methods, CBCT and IOS, were used to evaluate accuracy values.

Regarding the mean difference of the accuracy, no statistically significant difference was found in the two methods of data acquisition (post-operative CBCT or IOS impression) in the detection of apical deviation (*p* = 0.985), angular deviation (*p* = 0.979), depth (*p* = 0.754), and coronal deviation (Table 3).

## 5. Discussion

The accuracy of implant insertion by using static guides (Naviguide^®^), a dynamic navigation system (Navident™), or free hand was evaluated in this study by using two different methodologies, post-operative CBCT, or post-operative intra-oral scanner impression.

The validation of a new method (i.e., post-operative intra-oral scanner impression) to calculate deviations between planned and achieved implants is an essential point to reduce the biological risk for the patients and to ensure that the results with this last method could be compared to other data existing in the literature obtained with a post-operative CBCT. As the validation of IOS to assess implant accuracy was the first aim of the present study, the same model was printed 15 times. This choice could be pointed out as a possible cause of bias, but our results showed the absence of a statistically significant difference between the model by model dataset values.

There are two CAI methods available today, static (SCAI) and dynamic (DCAI).

SCAI uses a surgical guide that can be 3D printed using additive manufacturing (stereolithography), or drilled using subtractive manufacturing (numeric controlled machine) [22].

The “static” guide is a computer manufactured appliance based on the restoratively driven implant position on computer aided design (CAD) surgical software. Using a specific surgical kit, the drilling step and the implant placement are fully guided. Some recommendations are mandatory both in the planning and in the surgical phase. A drawback of this process is the inability to make changes once the stereolithographic guide has been manufactured [23].

The “dynamic” CAI option affords the surgeon freedom to make changes both during the planning and the surgical phase. In fact, dynamic techniques allow CBCT real time motion tracking of patient anatomy, drilling, and implant placing, thus allowing the real time visualization of the surgical treatment [24,25]. It is a computer guided free-hand technology that eliminates the need for computer-generated stereolithographic guides (SCAI) and direct visualization.

The literature confirms the accuracy of both methods.

Regarding the topic of accuracy in the clinical field, this procedure represents the possibility of complying with the planning and designing the prosthetic project before the surgery takes place. The chance to respect the horizontal orientation of the implant connection indeed, allows for the right orientation of the multi-unit abutment (MUA) usually placed on tilted implants to be known.

A different appraisal should be made about the free hand approach, also known as mental navigation. When a similar approach is followed, we would suggest frequently using a template outlining the final shape of the approved wax-up, in order to place the implant according to the prosthetic design. Even though the template would be opened at the occlusal level, resulting in a large hole with a great degree of freedom, with regard to compliance to the prosthetic proposal, the accuracy of implant placement can be easily obtained.

In terms of biologic soft tissue stability, the respect of leaving at least 1.5 mm of buccal and lingual bone from the implant platform should be mandatory, according to Monje et al. [26]. They compared implant sites in beagle dogs with residual buccal bone width smaller than 1.5 mm or bigger than 1.5 mm. They concluded that a thicker buccal bone wall (>1.5 mm) is exposed to less physiologic and pathologic bone loss compared with a tinner buccal bone wall (<1.5 mm).

As demonstrated in many studies, CAI approaches, compared to free hand ones, could ensure the respect of the bone volumes around implant heads, in terms of maintaining at least 1.5 mm of bone at both the buccal and lingual side [26,27,28].

It should be noted that guided approaches should not be confused with flapless surgery. The respect of the correct dimension of both hard and soft peri-implant tissues requires a pre-operative clinical and radiological analysis. The software used to plan the surgery really enables the surgeon to select the flap approach. Using a guided system, with these reported minimal deviations, it ensures the possibility to also perform, if needed, bone regeneration. Similarly, a flapped approach allows the soft tissue volume to easily increase, profiting by the second intention of wound healing.

Regarding the safety of the procedure, in the case of anatomic limits (i.e., nerves, roots, vessels), only a guided approach can be considered safe, especially if compared with a free hand procedure.

Our results showed a statistically significant difference for all the involved variables when the free hand approach, even if template guided, was compared to both the DCAI and SCAI approaches.

Concerning the static guides, Vercruyssen et al. [27] reported on a randomized, prospective study comparing the accuracy obtained when placing implants using static guidance (Materialise Universal R, Facilitate TM) with that obtained from free-hand (“mental navigation”), and pilot-drill templates in 72 fully edentulous jaws. The mean deviations (SD) for those implants placed with static guidance were 1.4 mm (0.7) at the entry point, 1.6 (0.7) mm at the apex, and 3.0 (2.0) degrees from the angular standpoint. In comparison, mean deviations (SD) measured with free-hand (“mental navigation”/unguided) were: 2.8 (1.5) mm at entry point, 2.9 (1.5) mm at the apex, and 9.9° (6.0°) for angular deviations. The above paper demonstrated the significant difference in deviation encountered when comparing static guidance to both pilot guided and free hand, confirming superior accuracy with static guidance. The results of this study should make us reflect on the acceptability of a similar degree of mean deviation: 2.8 mm at the entry point, 2.9 mm at the apex point, or almost 10 degrees of 3D deviation, in certain cases, representing an unacceptable result or a failure.

Tahmaseb et al. [28], in a recent systematic review and meta-analysis on fully and partially edentulous cases treated with a SCAI approach, reported mean deviations at a coronal point of 1.3 mm and 0.9 mm, respectively. At the apical position, the respective mean deviations were 1.5 mm and 1.2 mm. An angular deviation of 3.3 degrees was reported for both cases. The deviations, compared with those reported with a totally free hand approach (mental navigation), must be kept in mind, especially for advanced cases.

Multiple studies [29,30,31,32] have evaluated the accuracy of dynamic navigation systems and reported an in vitro accuracy of 1 to 2 mm when using first generation dynamic navigation systems.

Somogyi-Ganss (2014) [33] performed 80 in vitro osteotomies by using the dynamic navigation system. They respectively reported 1.14 mm, 1.71 mm and 2.99° for mean entry, apical, and angular points. Wagner (2003) [34] inserted 32 implants in five patients and described an angular accuracy of 6.4° with a range of 0.4 to 13.3.

Block et al. (2017) [35] reported on the placement accuracy obtained by three surgeons using a second-generation navigation system (X-Guide, X-Nav Technologies) to treat 100 patients. The deviations were also compared with freehand placement accuracy. Only partially edentulous cases were included, since a minimum of three adjacent teeth was required to hold a special clip enabling the navigation. The mean (SD) deviations with X-Guide were 0.87 (0.42) mm at entry (lateral/2D), 1.56 (0.69) mm at the apex (3D), and 3.62 (2.73) as the angular value. The unguided deviations had corresponding results (SD) of 1.15 (0.59) mm, 2.51 (0.86) mm, and 7.69 (4.92). No statistically significant differences were observed in the navigated placement between individual surgeons.

Jorba-Garcia et al. [36] in 2019 in vitro inserted 36 implants, 18 free-hand, and 18 by using a dynamic navigation system. They reported a significantly higher accuracy, especially the angular deviation for all the variables studied except for 3D entry and apex depth by using the dynamic navigation system. In fact, the deviations using a DCAI system were 1.29 mm at the 3D entry point, 0.85 mm at the 2D entry point, 1.32 mm at the 3D apex, 0.88 mm at the apex vertical, and 1.6 degrees as the angular deviation, while using the free-hand approach, they reported a deviation of 1.5 mm at the 3D entry, 1.26 mm at the 2D entry, 2.26 mm at the 3D apex, 0.57 mm at the apex vertical, and 9.7 degrees as the angular deviation.

Pellegrino et al. [37] treated 10 patients and 18 implants were placed using ImplaNav Navigational technology. They reported mean deviation values of 1.04 ± 0.47 mm at the entry point, 1.35 ± 0.56 mm at the apex, 0.43 ± 0.34 mm of depth deviation, and 6.46 ± 3.95 degrees of angular deviation.

Stefanelli et al. [38] placed 231 implants (89 arches) using Navident (Claronav, Toronto, ON, Canada) and reported a mean (SD) deviation of 0.71 mm (0.4) at the entry point, 1 mm (0.49) at the apex, and a mean angular discrepancy of 2.26 degrees (1.62°).

Stefanelli et al. [39], in a retrospective observational in vivo study, validated the accuracy of the trace and place method, a new digital way to perform a dynamic guided surgery without the need of any radiological thermoplastic stent. On 136 implants placed, they reported an overall mean deviation of 0.67 mm at the entry point level, 0.9 mm at the apex level, and 0.55 mm in depth, with an angular deviation of 2.5 degrees.

Aydemir and Arisan [40] inserted 86 implants in 30 patients in a split mouth study (free-hand vs. dynamic navigation system) and they reported the following deviation: 1.7 mm at the shoulder, 2.51 mm at the tip, and 10.04 degrees as angular error for free-hand, and 1.01 mm at shoulder, 1.83 mm at apex, and 5.59 degrees as the angular error with the use of a dynamic navigation system.

Nowadays, almost all of the studies have used the post-operative CBCT to evaluate implant deviations by overlapping the post-operative CBCT to the pre-operative one with the planning and calculating the resulting deviations between planned implants and inserted ones.

The method used to assess implant accuracy in the above-mentioned studies requires a second radiological scan of the patient, with an additional radiological exposition representing a biological risk that should be avoided if its only reason is the implant accuracy assessment. Other methods to assess implant deviations were proposed during the past years.

Tang et al. [20] used a gypsum cast and an external optical scanner to evaluate the accuracy of 32 implants and compared this method with the post-operative CBCT. The mean deviation between the digital registration method and the radiographic method was −0.03 ± 0.38 mm at the entrance point, −0.03 ± 0.57 mm at the apical point, and 0.60 ± 2.94 degrees as the angular discrepancy. No significant differences (*p* > 0.05) were found between the two methods.

Platzer et al. [41], in a study of 2013, used a laser scanner to calculate the deviation between pre-surgical and post-surgical casts.

Nickenig and coworkers [42] in 2010 suggested performing post-op 3D X-ray of the plaster casts with implant replicas inserted (the same cast used for definitive prosthesis manufacturing). In the study, they reported an average precision of implant position within 0.9 mm and 4.2° of implant axis deviation. They concluded their paper by suggesting the use of this method as an alternative to the validated CBCT matching.

Even if the above-mentioned last three studies suggested an alternative to post-op CBCT, not one of these studies verified if these methods produced comparable results to ones validated for post-op CBCT. This is an important key because the accuracy of new techniques/procedures needs to be compared to the ones existing in the literature.

Regardless of the guided technique used, static or dynamic, a limit of the validated CBCT matching method was pointed out in an interesting study published by Pettersson et al. in 2012 [43]. Of all the 139 placed implants, only 90 implants complied with the study methodology as the remaining 49 were affected by movements of the patients during preoperative and/or postoperative CBCT, making stronger and less precise the step of the implant shape (geometric image) reconstruction. The metric analysis was performed at the hex, apex, angle, and depth level, and it revealed statistically significant differences (*p* < 0.05) between planned and inserted implants in all four outcome variables.

Komiyama et al. [44], in a study published in 2011, using the same sample of patients and implants of Pettersson [43], made a pre-surgical gypsum cast based on each individual surgical guide and a post-operative cast by an intraoral conventional impression. Both casts were scanned by a probe scanner to evaluate the discrepancies of implant positions (pre-op and post-op). The interesting results were the reduction of the inaccuracies of CBCT itself, just as performed with a different method (i.e., the probe scanner) not affected by movements and radiological artifacts. Authors concluded suggesting the adoption of the model matching instead of the CBCT method to perform the accuracy because of its higher trueness and for the reduction in radiation for patients.

Skjerven et al. [19], in a study in 2019, proposed the use of the intra-oral scan with IOS after implant insertion to calculate implant 3D positions, in addition to the conventional post-op CBCT. A double comparison between deviations measured by pre-op and post-op CBCT and by pre-op CBCT and post-op IOS STL was performed. They reported that the difference in angular deviation between CBCT and IO scanning at the coronal point was −0.011 degrees (±0.6), the 3D deviation was 0.03 mm (±0.17), the distal deviation was 0.01 mm (±0.16), the vestibular deviation was 0.033 mm (±0.16), and the apical deviation difference was 0.09 mm (±0.16).

At the apical point, the 3D deviation was 0.04 mm (±0.22), the distal deviation was 0.06 mm (±0.19), the vestibular deviation was 0.032 mm (±0.23), and the apical deviation was 0.09 (±0.16) mm.

The differences between CBCT and IOS at the apical point were: 0.04 mm (±0.22) as 3D deviation, 0.06 mm (±0.19) as distal deviation, 0.032 mm (±0.23) as vestibular deviation, and 0.09 (±0.16) mm as apical deviation.

The authors concluded that IOS could replace post-op CBCT even if the deviations at the entry and apex points were found statistically significant; this was probably due to the limited number of implants (28 implants).

Monaco and coworkers [45], in 2019, performed a retrospective clinical study on the 2D/3D accuracies of implant position using four CAI systems. The measurements were based on the match of the IOS STL files with the pre-operative DICOM CBCT files. They documented the same surgical approach, three different ways to treat partially edentulous cases, where the guides were teeth supported, and only one way to treat fully edentulous gingival supported cases. Implant seating ranged from free hand to fully guided. All the full arch gingival cases received a flapless approach while all the previous were flapped. The results of their study showed no significant 3D positional differences among the four groups. For all implants, the mean differences at the implant head were 0.978 mm ± 0.476 mm and 1.20 mm ± 0.51 mm at the implant apex, with a mean angular deviation of 3.31° ± 1.99. They concluded that:All the guided dental supported surgeries were more accurate than those using a gingival support;The guided surgery is valid to reach an optimal implant placement;The use of IOS to compare final placement to implant planning is remarkable because of its accuracy and biological respect.

In this study, three methodologies were used (free-hand, surgical guide, and dynamic navigation system) for implant insertion and it was found, as by other authors in the literature, that the use of CAI (i.e., static or dynamic) represents an advantage for the clinician in terms of accuracy versus the free-hand insertion of implants.

The greatest innovation of this study was the method used to calculate implant accuracy.

The additional use of an intra oral scanner to assess the accuracy of placed implants compared to planned implants on a sample of 90 in vitro implants was investigated. The values found were similar and no statistically significant differences were reported for any deviation. These findings suggest that the use of the intra oral scanner can be considered as a new way to calculate the implant accuracy of partially edentulous cases because it is less invasive, more precise, and does not require additional appointments and treatments for the patients.

The scan body used to reveal the spatial implant position, after scanning and CAD elaboration, allowed us to obtain the results even if the transmucosal area could not be detected by the IOS. This is an issue related with the development of the transmucosal surface of the prosthetic framework.

The desire to use this method justified the replication of the same model and the results, in terms of the examiner’s performance, were satisfactory. No statistically significant differences from the first to the last model of the same examiner were noticed.

The study limitation is due to the fact that it is an in vitro study and that a single model was used.

## 6. Conclusions

Nowadays, a predictable result in terms of safety and fit with the prosthetic planning should be recommended. Many studies have demonstrated that a free hand approach is a significative cause of deviation from the surgical planning, being the main reason of prosthetic compromises with a great risk of affecting the safety of the surgery.

This study, with the in vitro limitation, confirmed that the use of one of the described CAI systems for implant surgery could perform a safer and more accurate implant insertion.

The second relevant conclusion is that, to date, to assess implant deviations, the matching of a pre-op and a post-op CBCT is the validated, most used, and published method. However, this second CBCT is taken to solely perform this evaluation, exposing the patient to a questionable biological risk.

The use of the IOS impression, instead of a post-operative CBCT for the implant accuracy evaluation, represents a resource that clinicians need to consider and investigate when planning a study on implant accuracy, in order to avoid the biologic impact of a second CBCT.

## Figures and Tables

**Figure 1 ijerph-17-09358-f001:**
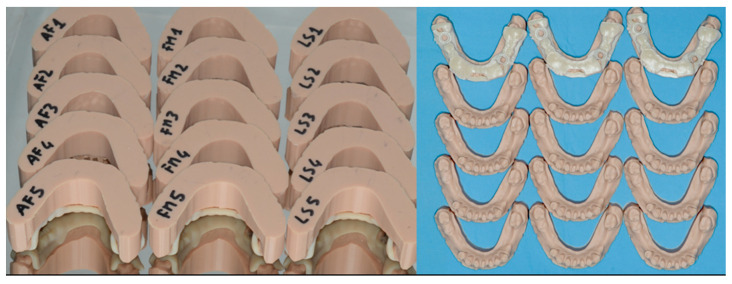
Fifteen marked identical resin models and the three surgical guides were manufactured to perform the study. Each model was coded according to randomization.

**Figure 2 ijerph-17-09358-f002:**
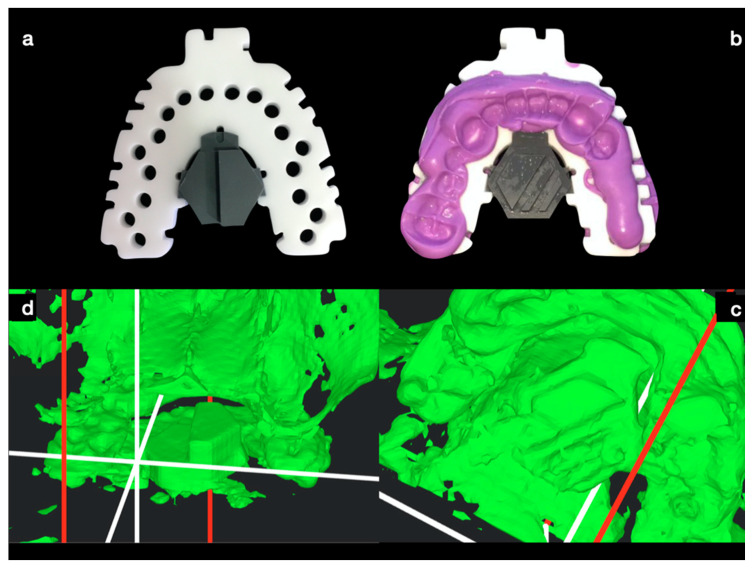
The fiducial marker device (NaviBite^®^, Biomax Spa, Italy) is helpful to align the STL files of the soft tissues and teeth with their own representation on DICOM files (**a**) and is customized using a silicon index material on both sides (**b**). A radiopaque marker is embedded in the center part of the device to be used, on both sides, for the STL-DICOM merging step, as shown in the 3D views (**c**,**d**).

**Figure 3 ijerph-17-09358-f003:**
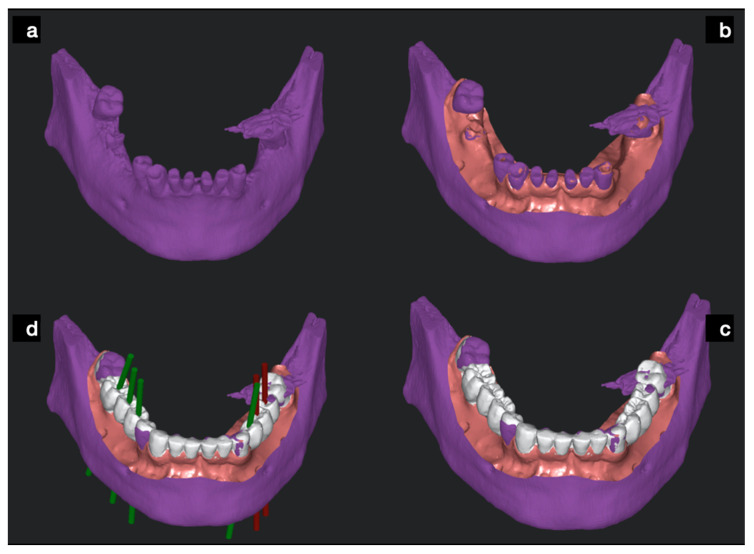
The bone-teeth cone beam computed tomography (CBCT) baseline status (**a**) with merged STL files of soft tissues (**b**) and with the wax-up scan (**c**) was taken with an extra oral scanner and loaded into the surgical SCAI software to be used for prosthetic driven implant planning (**d**).

**Figure 4 ijerph-17-09358-f004:**
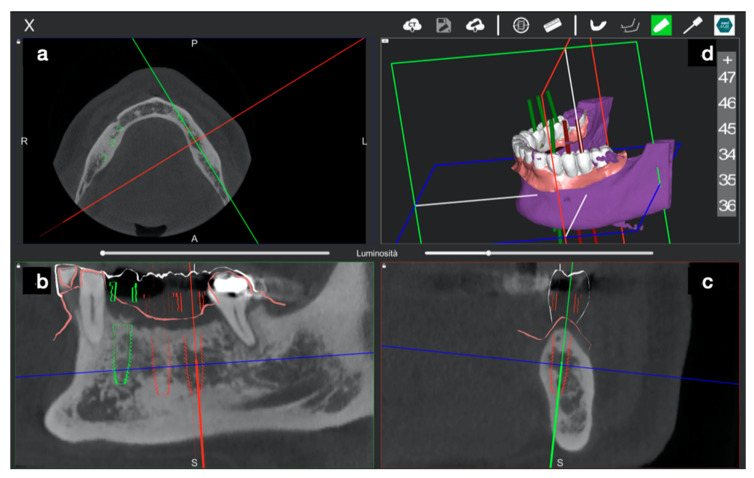
The planning software allows bone anatomy to be checked by three 2D windows (**a**–**c**) and one 3D window (**d**). At each window, the view is adjustable in order to better represent the bone section in the other windows. The final implant setup is performed according to the standard guidelines on 2-dimensional views.

**Figure 5 ijerph-17-09358-f005:**
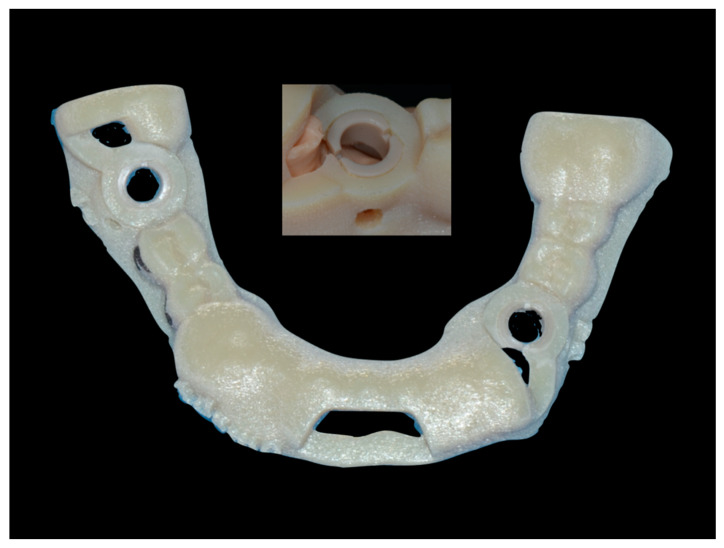
The stereolithographic surgical guide with PEEK sleeve embedded. It is important to note sleeves placed in the sites where SCAI protocol was planned and the wax-up shape of the guide for the remaining implant site.

**Figure 6 ijerph-17-09358-f006:**
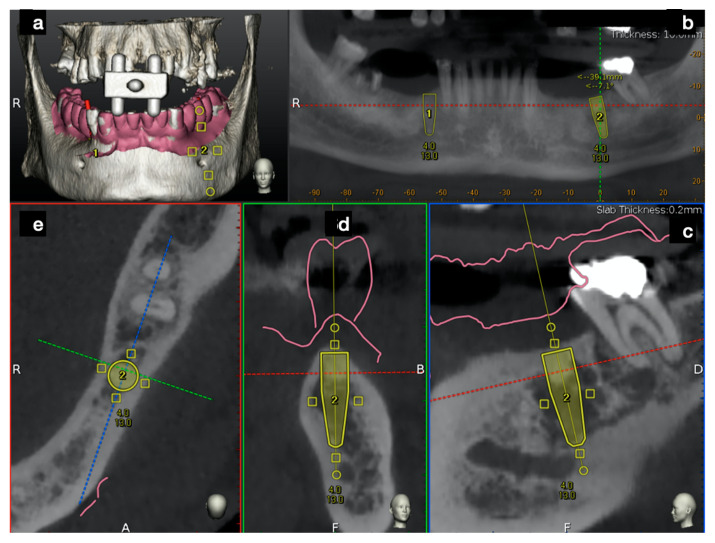
The Navident desktop was divided into five windows. The 3D view shows the fiducial marker used and the overlapped STL file taken from an extra oral scan of the lower jaw (**a**). Four 2D windows (panorex (**b**), sagittal (**c**), cross-sectional (**d**), and axial (**e**)) allowed us to perform the implant planning. After wax-up, the STL file was overlapped above the original patient’s DICOM data, and a prosthetic driven implant planning was performed.

**Figure 7 ijerph-17-09358-f007:**
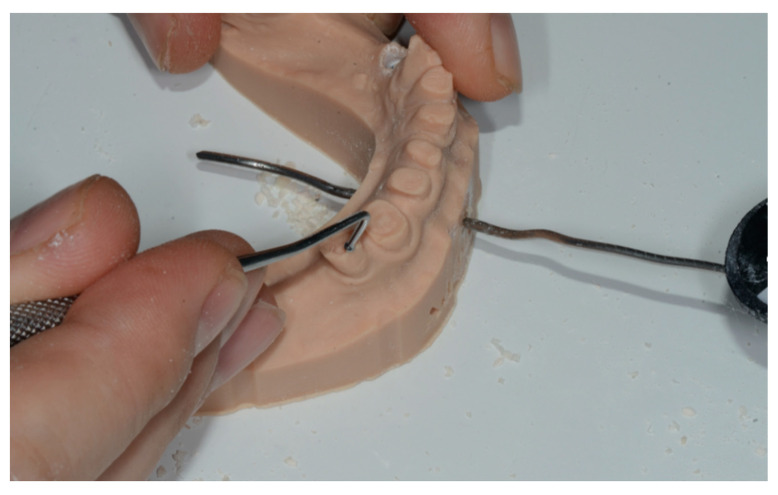
The tracer tool slides above the teeth for a 15 cm path to perform the required accuracy check.

**Figure 8 ijerph-17-09358-f008:**
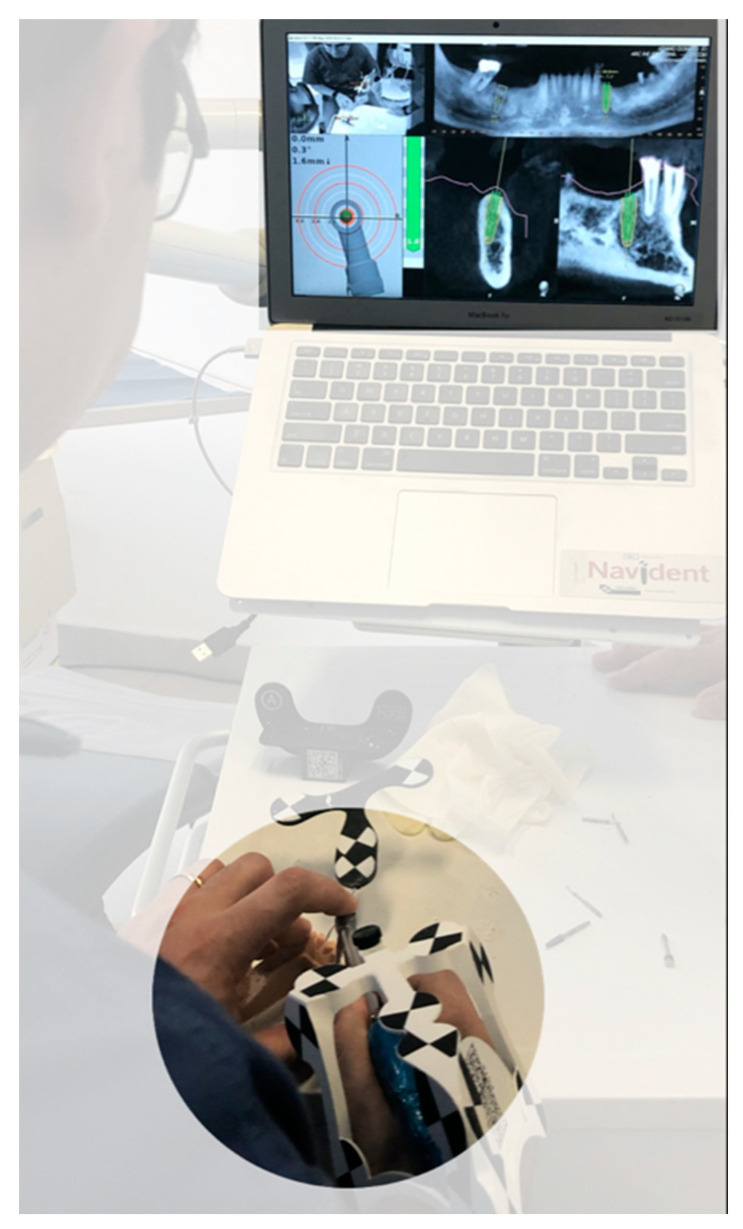
The unconventional way to prepare implant sites, following both the planned implant site and the drill advancement by an indirect screen view.

**Figure 9 ijerph-17-09358-f009:**
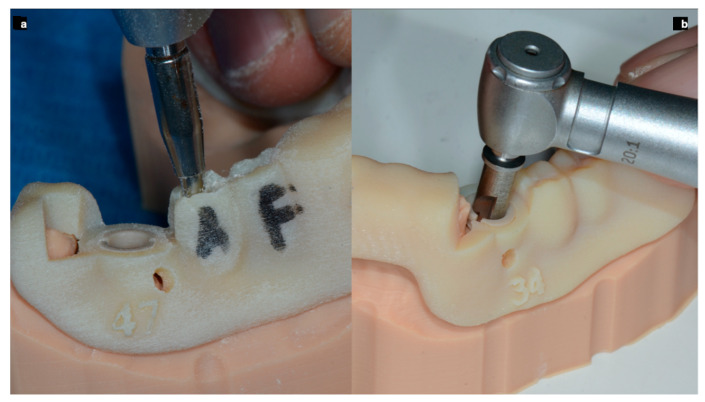
To perform the free hand (template guided) surgeries in the most accurate manner, the guides were perforated in the center of the occlusal plate of each selected site (**a**) and an attempt to be compliant with the project was done. Picture (**b**) shows a site with a sleeve and a dedicated bur performing the static SCAI approach.

**Figure 10 ijerph-17-09358-f010:**
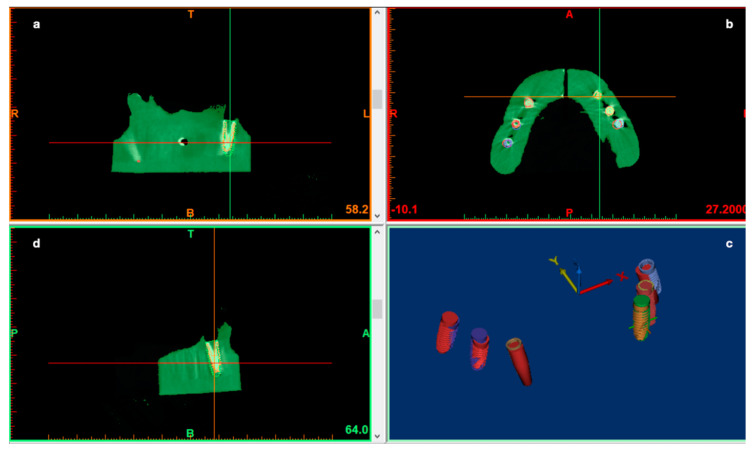
The pre-op planning and post-op CBCT of the model overlapped to analyze implant deviations. A preliminary segmentation of both implants and jaws was performed (**a**–**d**). The 3D objects of the jaws were hidden, leaving only 3D implants of both the pre-op and post-op scan (**c**). These implant positions and images were used to perform the deviation analysis.

**Figure 11 ijerph-17-09358-f011:**
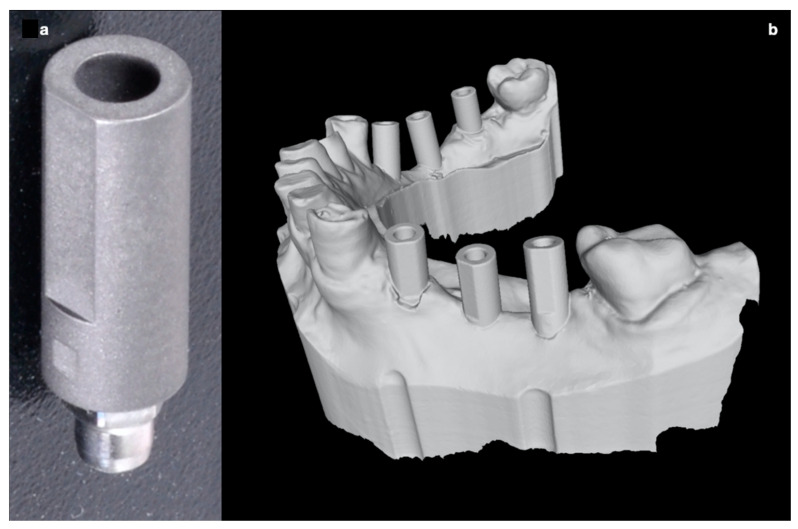
The position of the achieved implants was carried out by scanning the original scan abutment (**a**) for digital workflow (Biomax spa, Vicenza, VI, Italy). The picture (**b**) shows the resulting 3D digital cast obtained by the IOS of the resin model used to perform the surgeries.

**Figure 12 ijerph-17-09358-f012:**
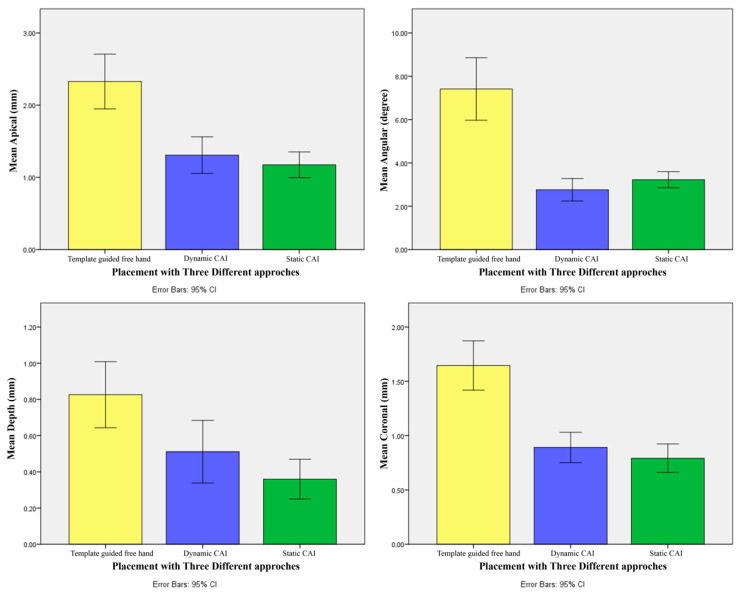
Simple bar charts (with confidence intervals) show the mean at coronal (mm), apex (mm), depth (mm), and angle (°) level among the three different approaches of implant placement.

**Figure 13 ijerph-17-09358-f013:**
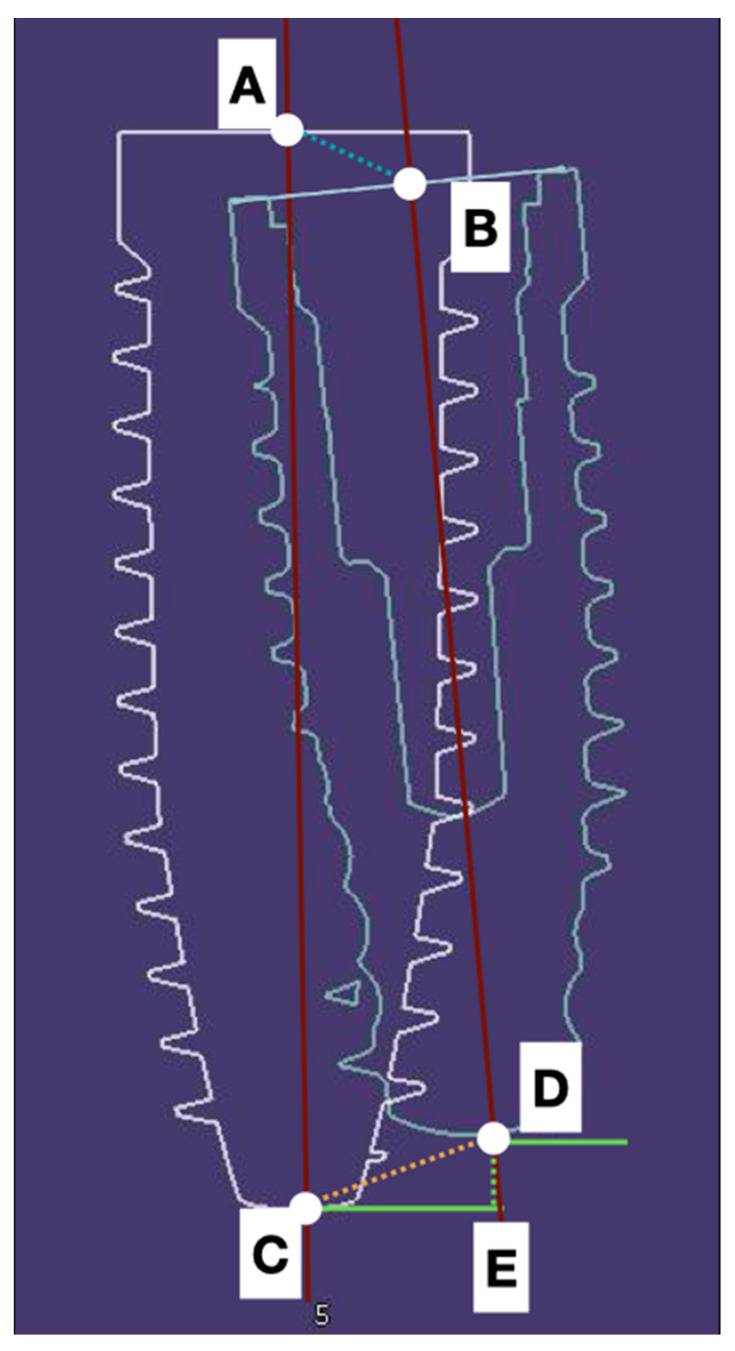
The reference points taken into account to assess deviations were the coronal (**A**,**B**) and the apical (**C**,**D**) deviation of the center part of the implant. The depth displacement (**D**,**E**) represented the final effect of the deviation on the final vertical position of the implant apex. The angular (**AC**,**BD**) deviations were also estimated.

**Table 1 ijerph-17-09358-t001:** The table reports the data, in terms of accuracy, by using the three different techniques for the total of 90 implants inserted (30 SG, 30 ND, 30 FH). The reported accuracy values were assessed by pre-op and post-op cone beam computed tomography (CBCT) superimposition.

System Used	Coronal (SD) mm	Angular (SD) Degree	Apical (SD) mm	Depth (SD) mm
SG	0.79 ± 0.35	3.23 ± 1.00	1.17 ± 0.48	0.36 ± 0.29
ND	0.89 ± 0.37	2.76 ± 1.38	1.31 ± 0.68	0.51 ± 0.46
FH	1.65 ± 0.61	7.41 ± 3.87	2.33 ± 1.01	0.83 ± 0.49

**Table 2 ijerph-17-09358-t002:** Difference between the means of implant inserted by using the dynamic navigation system and surgical guide. The reported results were assessed by pre-op and post-op CBCT superimposition.

Multiple Comparisons							
Dependent Variable	(I) Placement with Different Approaches	(J) Placement with Different Approaches	Mean Difference (I-J)	Std. Error	Sig.	95% Confidence Interval	
						Lower Bound	Upper Bound
**Apical (mm)**	Template guided free hand	Dynamic CAI	1.01900 *	0.22265	0.000	0.4814	1.5566
		Static CAI	1.15333 *	0.20461	0.000	0.6559	1.6508
	Dynamic CAI	Template guided free hand	−1.01900 *	0.22265	0.000	−1.5566	−0.4814
		Static CAI	0.13433	0.15128	0.650	−0.2306	0.4993
	Static CAI	Template guided free hand	−1.15333 *	0.20461	0.000	−1.6508	−0.6559
		Dynamic CAI	−0.13433	0.15128	0.650	−0.4993	0.2306
**Angular (degree)**	Template guided free hand	Dynamic CAI	4.65267 *	0.74930	0.000	2.8218	6.4835
		Static CAI	4.18867 *	0.72864	0.000	2.4003	5.9770
	Dynamic CAI	Template guided free hand	−4.65267 *	0.74930	0.000	−6.4835	−2.8218
		Static CAI	−0.46400	0.31109	0.303	−1.2142	0.2862
	Static CAI	Template guided free hand	−4.18867 *	0.72864	0.000	−5.9770	−2.4003
		Dynamic CAI	0.46400	0.31109	0.303	−0.2862	1.2142
**Depth (mm)**	Template guided free hand	Dynamic CAI	0.31500 *	0.12291	0.034	0.0193	0.6107
		Static CAI	0.46600 *	0.10428	0.000	0.2137	0.7183
	Dynamic CAI	Template guided free hand	−0.31500 *	0.12291	0.034	−0.6107	−0.0193
		Static CAI	0.15100	0.10016	0.296	−0.0910	0.3930
	Static CAI	Template guided free hand	−0.46600 *	0.10428	0.000	−0.7183	−0.2137
		Dynamic CAI	−0.15100	0.10016	0.296	−0.3930	0.0910
**Coronal (mm)**	Template guided free hand	Dynamic CAI	0.75500 *	0.13008	0.000	0.4404	1.0696
		Static CAI	0.85367 *	0.12790	0.000	0.5440	1.1634
	Dynamic CAI	Template guided free hand	−0.75500 *	0.13008	0.000	−1.0696	−0.4404
		Static CAI	0.09867	0.09324	0.544	−0.1256	0.3230
	Static CAI	Template guided free hand	−0.85367 *	0.12790	0.000	−1.1634	−0.5440
		Dynamic CAI	−0.09867	0.09324	0.544	−0.3230	0.1256

* The mean difference is significant at the 0.05 level.

**Table 3 ijerph-17-09358-t003:** The table shows that there were no statistically significant differences between the means by using the post-operative CBCT or IOS impression to evaluate implant deviations.

	*t*-Test for Equality of Means				
	Mean Difference	Sig.	Std. Error Difference	95% Confidence Interval of the Difference	
				Lower	Upper
Apical	0.00256	0.985	0.13505	−0.26396	0.26907
Angular	−0.01267	0.979	0.47324	−0.94654	0.92121
Depth	−0.02178	0.754	0.06949	−0.15890	0.11535
Coronal deviation	0.03856	0.664	0.08863	−0.13635	0.21346

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
