# Peer review of "Validation of an Intra-Oral Scan Method Versus Cone Beam Computed Tomography Superimposition to Assess the Accuracy between Planned and Achieved Dental Implants: A Randomized In Vitro Study"

_ijerph, 2020, doi:10.3390/ijerph17249358_

Round 1

Reviewer 1 Report

Validation of an Intra-oral Scan method versus Cone Beam Computed Tomography superimposition to assess the accuracy between planned and achieved dental implants: a multi-center randomized in vitro study.

This experimental in vitro study examines a very interesting topic, since the exponential development of computer aided implantology (CAI) systems. The article is interesting and takes into account many crucial issues about CAI.

One of the points of strength of this article is the detailed materials and methods section, in which the authors describe very clearly every clinical step necessary to achieve correct clinical results, when planning a computer guided implant placement.

The authors also provided a very detailed discussion, with an accurate and interesting literature analysis. The paper is suitable for publication.

Nevertheless, some issues have to be highlighted:

  • English language, especially in Abstract and Introduction sections, is rather odd or incorrect. I kindly ask the authors to carefully analyze and reformulate the written sentences, in order to allow an easier understanding.
  • A factor which might change the accuracy of the intraoral scanning is that the in vitro model, due to its manufacturing, does not reproduce the transmucosal area, which is the most difficult to reproduce with a high level of accuracy. Do the authors believe that this factor could be relevant in case this study would take place in an in vivo setting? A brief comment should be included in the Discussion section.

Author Response

Validation of an Intra-oral Scan method versus Cone Beam Computed Tomography superimposition to assess the accuracy between planned and achieved dental implants: a multi-center randomized in vitro study.

This experimental in vitro study examines a very interesting topic, since the exponential development of computer aided implantology (CAI) systems. The article is interesting and takes into account many crucial issues about CAI.

One of the points of strength of this article is the detailed materials and methods section, in which the authors describe very clearly every clinical step necessary to achieve correct clinical results, when planning a computer guided implant placement.

The authors also provided a very detailed discussion, with an accurate and interesting literature analysis. The paper is suitable for publication.

Nevertheless, some issues have to be highlighted:

1.English language, especially in Abstract and Introduction sections, is rather odd or incorrect. I kindly ask the authors to carefully analyze and reformulate the written sentences, in order to allow an easier understanding. 

author reply:The abstract has been deeply revised both in its contents and in language

2. A factor which might change the accuracy of the intraoral scanning is that the in vitro model, due to its manufacturing, does not reproduce the transmucosal area, which is the most difficult to reproduce with a high level of accuracy. Do the authors believe that this factor could be relevant in case this study would take place in an in vivo setting? A brief comment should be included in the Discussion section. 

author reply:The method used is based on the scan of a physical object (scan body), regardless of the reproduction of the peri-implant mucosal area. Due to its known geometry, a CAD software could be used to determine the implant position, that is the studied parameter. Moreover the best fit alignment has been performed by matching the residual teeth of both spans. The reported protocol has been used for partially edentulous cases. The fully edentulous cases should require more reference point because the mucosal area could affect the results. A brief comment about this point has been added in the discussion section

Reviewer 2 Report

I appreciate the opportunity to review the article entitled “Validation of an Intra-oral Scan method versus Cone Beam Computed Tomography superimposition to assess the accuracy between planned and achieved dental implants: a multi-center randomized in vitro study”. Intra-oral Scan is interested as an evaluated method for the placed dental implants instead of CBCT.  This research may well be of reference to implant surgeons. I put some points I have noticed below at this time.  Hope this helps.

  1. The title of this manuscript includes the terms “ a multi-center randomized in vitro study”.  The reason for the terms could not be found out, although the authors belong to different institutes. Its ground should be described in Materials and Methods.
  2. The number of people who performed in vitro surgery should be described in Materials and Methods, although who did the surgery is shown in “ Author contribution”.  Are they implant specialists?
  3. I guess that the data of Table 1 and Table 2 were produced with a comparison between pre-?op and post-?op CBCT images.  This point should be indicated obviously.
  4. In Table 2, it is difficult to read the part where the characters are out of alignment.
  5. The characters in Figure 12 are too small to read.
  6. The legends of Figure 10 and 13 should be detailed.
  7. “Discussion” should be rewritten significantly. Most of the part are just the introduction of past researches. You should compactly discuss focusing on your own data.

Author Response

I appreciate the opportunity to review the article entitled “Validation of an Intra-oral Scan method versus Cone Beam Computed Tomography superimposition to assess the accuracy between planned and achieved dental implants: a multi-center randomized in vitro study”. Intra-oral Scan is interested as an evaluated method for the placed dental implants instead of CBCT.  This research may well be of reference to implant surgeons. I put some points I have noticed below at this time.  Hope this helps.

1.The title of this manuscript includes the terms “ a multi-center randomized in vitro study”.  The reason for the terms could not be found out, although the authors belong to different institutes. Its ground should be described in Materials and Methods

authors reply:According to your suggestion we decided to not consider this study multi-centered because, even if three operators, of different institutes, performed the surgeries, they operated at the same time in the same office with the same devices

2.The number of people who performed in vitro surgery should be described in Materials and Methods, although who did the surgery is shown in “ Author contribution”.  Are they implant specialists? 

author reply:This point has been discussed in M&M section to understand operators’ composition and expertise.

3.I guess that the data of Table 1 and Table 2 were produced with a comparison between pre-?op and post-?op CBCT images.  This point should be indicated obviously. 

auhtor reply: It’s right and it has been indicated in the table legend.

4.In Table 2, it is difficult to read the part where the characters are out of alignment. 

author reply: The table has been enlarged for an easier reading

5.The characters in Figure 12 are too small to read. 

authors reply: The characters dimension has been increased.

6. The legends of Figure 10 and 13 should be detailed. 

author reply: Both legends and figures have been remaked to fit with your request

7. “Discussion” should be rewritten significantly. Most of the part are just the introduction of past researches. You should compactly discuss focusing on your own data. 

author reply: According to the original order we wrote, the discussion has been reviewed following your suggestions.

Reviewer 3 Report

Although this article might be of interest to the readers, I think the content and quality of presentation can be improved. The methods and conclusion sections can be more clear and more relevant to the topic.

Author Response

Although this article might be of interest to the readers, I think the content and quality of presentation can be improved. The methods and conclusion sections can be more clear and more relevant to the topic.

author reply: All the manuscript’s sections have been reviewed.

Round 2

Reviewer 2 Report

Thank you for submitting a revised version. The points I put in the last review have been improved. The "Discussion" is still verbose, but i think it's acceptable.

This manuscript is a resubmission of an earlier submission. The following is a list of the peer review reports and author responses from that submission.

Round 1

Reviewer 1 Report

language errors 

Abbreviations should clearly state for the first when it present in the text.

Multiple font sizes were used by authors 

References were not prepared according to the journal format

Figure 4 and Figure 5  not been required.

Figure 9 focus on the screenshot from the computer or else reorganize it.

Table 2 appeared first in-text than Table 1.

Italics were used in table 1 and 2 which may not be required.

The methodology was good however, it should be sequenced in the proper format.

Results are appropriate, it is the better to state  objective-based since it very confusing for the readers in the present state  

Discussion, Previous studies used were mentioned in an orderly manner

References were stated in various styles 

Kindly go through the guidelines prior to submission.

Discussion on the clinical significance of such methods improves paper quality.

Author Response

Dear Reviewer,

following in the text your comments and our reply.

Kind regards.

--------

language errors: the whole text has been checked again.

Abbreviations should clearly state for the first when it present in the text. The abbreviations have been checked and corrected.

Multiple font sizes were used by authors: Font types have been definitely reformatted as requested in the guidelines for authors.

References were not prepared according to the journal format: References have been checked and reformatted in the ACS style guide as suggested in the guidelines for authors

Figure 4 and Figure 5 not been required. Figures 4-5 have been deleted, they had been inserted only to complete the description.

Figure 9 focus on the screenshot from the computer or else reorganize it. The idea of this picture was to emphasize the importance of the tags placed both on the handpiece and the patient’s jaw to allow the device to make a real time triangulation between the drill and the patient anatomy. It makes possible to follow in real time the osteotomy into several windows on the screen. To look directly into the screen instead of the patient’s mouth represents one of the most important differences with Static CAI.

Table 2 appeared first in-text than Table 1. Table 1 has been repositioned in the right place

Italics were used in table 1 and 2 which may not be required. Table 1 didn’t include Italics, Table 2 has been corrected.

The methodology was good however, it should be sequenced in the proper format. M&M description has been corrected in order to be more friendly to read and to follow a clearer scheme. In the first section we wrote a general presentation of approaches and methods used to carry out the analysis (line 89 to line 114). Then three sections (2.1, 2.2 and 2.3) made a deeper clarification of the three approaches used (from line 118 to line 278), followed by the last 2.4 section (from line 279 to line 312) where the methodology of data collection was described.

Results are appropriate, it is the better to state objective-based since it very confusing for the readers in the present state. Results section has been expanded including data description for each method of implant placement and each assessment of accuracy method (from line 330 to line 350).

Discussion, Previous studies used were mentioned in an orderly manner. The discussion has been reorganized with a general description (from line 367 to line 397), a part dedicated to some references about the topic of the study, both SCAI and DCAI (from line 399 to line 462), a critical comment about CBCT study (from line 463 to 468), alternative proposals to CBCT method (from line 469 to line 482), a comment about the need of validation of new proposals (483 to 488), a citation of two essential studies where the accuracy of the same sample of implants has been measured with two different methods, demonstrating the limits of CBCT method (489 to 509), two similar proposals to the ones we are presenting (510 to 544), our considerations related to the results of the present study (545 to 561)

References were stated in various styles References have been checked and reformatted in the ACS style guide as suggested in the guidelines for authors

Kindly go through the guidelines prior to submission. The paper has been reviewed following guidelines.

Discussion on the clinical significance of such methods improves paper quality. The discussion has been reorganized with a general description (from line 367 to line 397), a part dedicated to some references about the topic of the study, both SCAI and DCAI (from line 399 to line 462), a critical comment about CBCT study (from line 463 to 468), alternative proposals to CBCT method (from line 469 to line 482), a comment about the need of validation of new proposals (483 to 488), a citation of two essential studies where the accuracy of the same sample of implants has been measured with two different methods, demonstrating the limits of CBCT method (489 to 509), two similar proposals to the ones we are presenting (510 to 544), our considerations related to the results of the present study (545 to 561)

Reviewer 2 Report

Suggestions for Authors,

In general the manuscript is well written and each section is well organized. The described methods are current, however before the acceptance the revision is required.

  1. The abbreviation "CBCT" should be explain both within the title and the abstract. 
  2. The Table 1 should be before Table 2 within the manuscript - it should be corrected.
  3. I suggest to add the subdivision of the figures as example 1A,B...etc. (some of them are not clear).
  4. All the references should be checked carefully once again. Please use the Author guidelines.

Author Response

Dear Reviewer,

following in the text your comments and our reply.

Kind regards.

-----------

In general the manuscript is well written and each section is well organized. The described methods are current, however before the acceptance the revision is required.

  1. The abbreviation "CBCT" should be explain both within the title and the abstract. All the abbreviation has been written extended (for the first time)
  2. The Table 1 should be before Table 2 within the manuscript - it should be corrected. Table 1 and 2 have been correctly positionated within the text.
  3. I suggest to add the subdivision of the figures as example 1A,B...etc. (some of them are not clear). Figures 1,2,3,5,9,10,11 have been subdivided and a description has been added to the legend of each figure.
  4. All the references should be checked carefully once again. Please use the Author guidelines. References have been checked and reformatted in the ACS style guide as suggested in the guidelines for authors

Reviewer 3 Report

This is not a study, or research protocol; rather, is a description of a commericial product, compared with other products/techniques.

Nevertheless, also assuming that this is a study protocol, it is completely out of the main (and also collateral) journal's scope. A dental journal is undoubtedly more fitting for this article.

Finally, I do the use of self-citations: this is a scientific misconduct and should not be used to improve the scientific profile.

For these reasons, and for the poor scientific soundness of this article, I do suggest rejection with no further resubmission to this specific journal.

Author Response

Dear reviewer,

following our reply to your comments.

Kind regards

-----

This is not a study, or research protocol; rather, is a description of a commericial product, compared with other products/techniques. The object of this paper is the result of an in-vitro study where one SCAI brand, and one DCAI brand supported us giving all the needed devices and components. The scanner used for the accuracy assessment is property of one of the coauthors of the paper. The aim of this study was to assess if computer aided implantology represents an advantage for the clinicians in implant planning and placement compared with free hand approach. The second aim was to evaluate if another method (IOS post-op) was possible to calculate implant accuracy in order to minimize biological exposure to radiation. A lot of osteotomies and measurements have been carried out in a scientific approach. There was definitely no intention to promote any commercial device or brand.

Nevertheless, also assuming that this is a study protocol, it is completely out of the main (and also collateral) journal's scope. A dental journal is undoubtedly more fitting for this article. The paper has been submitted to a special issue of IJERPH titled as “Digital dentistry, Implantology and Maxillo-Facial Disease”, then it’s our opinion that a coherence with the aim of that issue can exist.

Finally, I do the use of self-citations: this is a scientific misconduct and should not be used to improve the scientific profile. Even if two articles could be forced to be cited in this paper we thank you for your constructive criticism and we decided to remove them from the references list. It actually contains only strongly related articles on the theme of the paper.

For these reasons, and for the poor scientific soundness of this article, I do suggest rejection with no further resubmission to this specific journal. We obviously ask you to reconsider your opinion after reading again the paper and after reading our previous rebuttals to your comments.

Round 2

Reviewer 1 Report

Manuscript looks promising 

A few typo errors need to be addressed